# The Immunology of DLBCL

**DOI:** 10.3390/cancers15030835

**Published:** 2023-01-29

**Authors:** Taishi Takahara, Shigeo Nakamura, Toyonori Tsuzuki, Akira Satou

**Affiliations:** 1Department of Surgical Pathology, Aichi Medical University Hospital, Nagakute 480-1195, Japan; 2Department of Pathology and Laboratory Medicine, Nagoya University Hospital, Nagoya 466-8550, Japan

**Keywords:** DLBCL, immune escape, immunosenescence, immune-privileged site

## Abstract

**Simple Summary:**

Diffuse large B-cell lymphoma (DLBCL) is the most common type of malignant lymphoid neoplasm and includes morphologically and molecularly heterogeneous disease subtypes. Genetic aberrations of tumor cells are strongly related to the signature of the tumor microenvironment. In this review, we summarize common genetic aberrations associated with immune escape, immune cell subs involved in DLBCL pathogenesis, and distinct microenvironmental signatures identified using next-generation sequencing and single-cell technologies. We also discuss the pathogenic role of immunosenescence in Epstein-Barr virus-positive DLBCL. Moreover, as DLBCL exhibits unique pathogenesis in immune-privileged sites, we also present a hypothetical model of DLBCL development in immune-privileged sites.

**Abstract:**

Diffuse large B-cell lymphoma (DLBCL) is an aggressive malignancy and is the most common type of malignant lymphoid neoplasm. While some DLBCLs exhibit strong cell-autonomous survival and proliferation activity, others depend on interactions with non-malignant cells for their survival and proliferation. Recent next-generation sequencing studies have linked these interactions with the molecular classification of DLBCL. For example, germinal center B-cell-like DLBCL tends to show strong associations with follicular T cells and epigenetic regulation of immune recognition molecules, whereas activated B-cell-like DLBCL shows frequent genetic aberrations affecting the class I major histocompatibility complex. Single-cell technologies have also provided detailed information about cell–cell interactions and the cell composition of the microenvironment of DLBCL. Aging-related immunological deterioration, i.e., immunosenescence, also plays an important role in DLBCL pathogenesis, especially in Epstein-Barr virus-positive DLBCL. Moreover, DLBCL in “immune-privileged sites”—where multiple immune-modulating mechanisms exist—shows unique biological features, including frequent down-regulation of immune recognition molecules and an immune-tolerogenic tumor microenvironment. These advances in understanding the immunology of DLBCL may contribute to the development of novel therapies targeting immune systems.

## 1. Introduction

Diffuse large B-cell lymphoma (DLBCL) is the most common type of malignant lymphoma, comprising around 40% of all malignant lymphoid neoplasms and including a morphologically and molecularly heterogeneous group of lymphomas [1,2]. Based on gene expression profiling, DLBCLs are subdivided into activated B-cell-like (ABC-DLBCL) and germinal center B-cell-like (GCB-DLBCL) types, with each group thought to represent a differentiation state [3]. This cell-of-origin (COO)-based classification is associated with clinical course and biological features, with ABC-DLBCL exhibiting a worse prognosis than GCB-DLBCL. ABC-DLBCL displays chronically active BCR signaling, resulting in constitutive NF-κB activity [4,5]. On the other hand, a subset of GCB-DLBCLs rely on the activation of PI3K signaling induced by tonic B-cell receptor (BCR) signaling [6]. ABC-DLBCL and GCB-DLBCL exhibit different distributions of genetic aberrations, and recently published papers have integrated genetic aberrations into the COO classification [7,8,9,10]. 

Recent advances in our understanding of the molecular pathogenesis of DLBCL have enabled the development of novel therapeutic strategies targeting intracellular signaling pathways [11]. Still, there remains a need to elucidate molecular aspects of the interactions between neoplastic cells and the tumor microenvironment (TME) and systemic immune system [12]. As novel immune system-targeting therapies have improved patient survival—including CAR-T therapy and treatment with immune checkpoint inhibitors—it would be useful to establish good prognostic markers to select patients who will benefit from these therapies. In this review, we discuss the current knowledge regarding the immune escape mechanisms of neoplastic cells of DLBCL, which play a major role in tumor progression and its association with host immune systems.

## 2. Genomic Aberrations of DLBCL Affecting Immune Status

In many solid neoplasms, the frequency of somatic mutations is associated with the “neoantigen burden”. Neoantigens are non-self peptides that originate from tumor-specific altered gene products. Compared with other solid malignancies, DLBCLs are characterized by somatic hypermutation (SHM) in genes encoding variable regions of the immunoglobulin gene (IGV) and other off-target genes, such as *PIM1*. DLBCLs with high levels of SHM in genes of the immunoglobulin heavy chain variable region reportedly harbor significantly higher numbers of Ig-derived neoantigens [13]. Some DLBCLs are infected by Epstein-Barr virus (EBV) and express EBV-derived antigen on the tumor cell surface [14,15]. These antigens generate anti-tumoral responses by antigen-specific T cells [16]; therefore, neoplastic B cells employ several immune escape mechanisms to evade surveillance by antigen-specific T cells and other immune cells. Indeed, approximately three-fourths of DLBCLs harbor genetic aberrations in genes associated with immune escape [7]. These immune escape-associated gene aberrations are enriched in the C1 and C5 subtypes or MCD genetic subtype, characterized by frequent presence of *MYD88^L265P^* and/or *CD79B* mutations and almost all of which are ascribed to ABC-DLBCL by the COO classification [7,8]. These lymphoma subtypes show high activity of an activation-induced cytidine deaminase (AID) footprint, which may produce high immunogenicity [7]. 

The most prevalent immune escape-associated mechanism is a lack of cell-surface expression of the class I major histocompatibility complex (MHC-I), which is observed in around 50% of DLBCLs [17]. MHC-I molecules are heterodimers that comprise a heavy (α) chain, encoded by *HLA-I*, and a β2-microglobulin (β2M) light chain, encoded by *B2M* [18]. Self and non-self peptides, including viral- and tumor-associated antigens, are degraded in the proteasome, and presented by the MHC-I complex at the cell surface. Antigen-specific cytotoxic T lymphocytes (CTL) interact with the MHC-I complex on target cells through a T-cell receptor (TCR) complex and, in the presence of a co-stimulatory signal, CTL are activated and kill the target cells [19]. Thus, MHC-I is required for CTL activation, and is downregulated in many types of malignancies to reduce recognition by CTLs [20]. Among DLBCLs, loss of MHC-I is preferentially found in mature B-cell malignancies, and 80% of MHC-I-negative DLBCLs harbor somatic inactivation of *B2M* and *HLA-I*. Genetic aberrations leading to MHC-I loss are relatively enriched in ABC-DLBCL compared to in GCB-DLBCL [21]. Biallelic genetic aberrations of *B2M* and *HLA-I* are mutually exclusive, suggesting that these genetic aberrations play complementary functions in MHC-I downregulation. The finding of higher mutation burden in DLBCLs lacking MHC-I, also highlights the role of immune escape in these tumors [17]. Interestingly, 70% of DLBCLs with MHC-I expression also harbor monoallelic *HLA-I* alterations, implying that reduced *HLA-I* gene diversity may impair tumor cells’ ability to present antigens to CTL by lowering the repertoire of functional MHC-I molecules. Although tumor cells lacking MHC-I can evade CTL, diminished MHC-I expression makes them vulnerable to natural killer (NK) cell-mediated cytotoxicity [21]. The MHC-I acts as the inhibitory ligand for NK cells and, under normal circumstances, MHC-I expression prevents NK cells from attacking host cells [22]. Loss of CD58 expression is another important mechanism of immune escape by altering immune recognition. Known as lymphocyte function-associated antigen 3 (LFA-3), CD58 is a natural ligand of CD2 expressed on T/NK cells. The CD2–CD58 interaction is an important component of immunological synapses, and CD58 expression on tumor cells is required both for NK cell and CTL-mediated cytolysis of DLBCL. Inhibiting CD58 results in diminished recognition and cytolysis of target cells by both CTLs and NK cells, and re-expression of CD58 induced a significant increase in NK cell-mediated cytolysis [23]. Indeed, approximately three-fourths of DLBCLs that lack MHC-I also lack CD58 expression [21], indicating that escape from both CTL and NK cells play an important role in DLBCL pathogenesis.

The immune escape mechanism also involves attenuated expression of class II major histocompatibility complex (MHC-II). MHC-I is generally constitutively expressed by all nucleated cells, and MHC-II is usually expressed on the cell surface of professional antigen-presenting cells, including B cells [24]. MHC-II expression is driven by the transcriptional master regulator class II transactivator (CIITA), which is necessary and sufficient for induction of a fully functional MHC-II pathway. Genetic aberrations causing MHC-II downregulation were first discovered in primary mediastinal large B-cell lymphoma (PMBL) by Gascoyne’s group and are regarded as the characteristic genetic aberrations of that peculiar disease [25,26]. Genetic aberrations affecting *CIITA* are also enriched in GCB-DLBCL, DLBCL with plasmacytic differentiation, and EBV-positive diffuse large B-cell lymphoma (EBV^+^DLBCL) [8,27,28]. Loss of MHC-II expression is associated with reduced T-cell infiltration and inferior prognosis [27,29]. In contrast with the MHC-I, tumor-specific antigens of B-cell lymphomas presented by MHC-II are recognized by CD4^+^ T cells [30]. Tumor killing by immune cells is generally ascribed to CD8^+^ CTL, while CD4^+^ T cells have been traditionally thought to function in cytokine production and to help CD8^+^ CTL exert cytotoxic functions. However, growing evidence suggests that a population of CD4^+^ T cells are capable of direct and indirect cytotoxicity, and this population has been termed CD4^+^ CTL [31,32,33,34]. Thus, MHC-II downregulation also plays an important role in the pathogenesis and immune escape of DLBCL. 

In addition to the above-mentioned immune recognition molecules, DLBCL also commonly exhibits disruption of the co-stimulatory and co-inhibitory molecule genes that modulate the immune responses among T cells and B cells. Monoallelic and biallelic inactivation of *CD70* is found in 14% of overall DLBCL cases, and this frequency reaches around 50% among DLBCL cases with *BCL6* translocation [7,8]. CD70 is a type-II membrane glycoprotein belonging to the tumor necrosis factor superfamily, also known as TNFSF7. CD70 expressed on B cells interacts with CD27 on T cells [35], and this CD70–CD27 axis activates cell survival signaling, enhances T-cell proliferation, and is thought to have antitumor effects [36]. Although these findings suggest that *CD70* has tumor suppressor capability, it has also been implied to play an oncogenic role based on the inferior prognosis of patients with DLBCL, exhibiting high CD70 expression [35]. The other TNF superfamily gene, *TNFSF9* (*CD137L*, *4-1BBL*), resides on chromosome 19p13.3 adjacent to *CD70*, and is frequently co-deleted with *CD70* [35,37]. TNFSF9 is expressed on antigen-presenting cells, including B cells, and binds to TNFRSF9 (CD137, 4-1BB) on activated T cells. Activation of the TNFSF9–TNFRSF9 pathway promotes T-cell proliferation through the regulation of cyclin-dependent kinases sustains the survival of activated T cells through expression of anti-apoptotic molecules and contributes to cytokine production [38]. The role of *TNFSF9* deletion in DLBCL pathogenesis remains unclear, but a 19p13.3 deletion involving *TNFSF9* was reported to be associated with adverse outcomes in DLBCL [39]. Inactivating gene mutation and genomic deletion of *tumor necrosis factor receptor superfamily member 14* (*TNFRSF14*, *HVEM*) are the characteristic genetic aberrations in EZB-type DLBCL and are associated with transformation of follicular lymphoma [8,40]. TNFRSF14 is broadly expressed on various types of cells within lymphoid tissue, including T cells, B cells, and dendritic cells (DCs). TNFRSF14 has several binding partners, including TNF superfamily member 14 (TNFSF14, LIGHT), CD160, and B- and T-lymphocyte attenuator (BTLA). Loss of those interactions leads to accumulation of follicular helper T cells (TFH) in the TME, inducing a tumor-supportive TME, and promoting resistance to CAR-T therapies [40]. The co-inhibitory molecule programmed cell death 1 ligand 1 (PD-L1, CD274) plays important roles in the development of many types of malignancies, including B-cell lymphoma [41]. PD-L1 on tumor cells binds to programmed cell death 1 (PD-1, PDCD1) on T cells, which is colocalized with the TCR, inhibiting T-cell activation and inducing T-cell exhaustion [42]. Another PD-1 binding partner, programmed cell death 1 ligand 2 (PD-L2), has a similar function to PD-L1, and both *PD-L1* and *PD-L2* reside on chromosome 9p24.1. [43]. Gain, amplification, and translocation of 9p24.1 all lead to overexpression of PD-L1 and PD-L2 on tumor cells [44]. Additionally, structural variations (SV) in the 3′-untranslated region (3′-UTR) of *PD-L1* upregulates *PD-L1* transcripts via mRNA stabilization [45]. SV in the 3′-UTR disrupt the C-terminus of *PD-L1*, such that the molecule is not detected by antibodies recognizing the intracellular domain [46]. These genomic alterations affect 10–20% of DLBCLs overall. PD-L1-positive DLBCL shows high cytotoxic activity of tumor-infiltrating T cells, suggesting that PD-L1 plays an important role in the immune escape of tumor cells, especially in cases with high immunogenicity [47,48]. Notably, Shimada et al. recently investigated intravascular large B-cell lymphoma (IVL) and found that 38% of cases harbored SV involving the 3′-UTR of *PD-L1/PD-L2*. This frequency was detected by liquid biopsy, i.e., plasma-derived cell-free DNA (cfDNA), and the issues related to this method are discussed later. In addition, several intracellular signaling pathways are known to induce PD-L1 expression [49]. For example, STAT3 binds to PD-L1 gene promoter and is required for PD-L1 gene expression [50]. The role of STAT3 activation in lymphoma has been highlighted mainly in the ABC DLBCL and EBV^+^DLBCL, and EBV oncoprotein LMP1 can induce STAT3 activation [51,52]. Notably, PD-L1 is not only expressed on tumor cells, but also on background inflammatory cells, such as tumor-associated macrophages (TAM) in the TME [53,54,55]. PD-L1-positive DLBCL was associated with inferior clinical outcome when treated with a rituximab-containing regimen, but showed clinical response to PD-1 blockade therapy, indicating that PD-1 can be a pharmacological target in this disease subtype [48,54]. 

Recently, Ennishi et al. discovered *TMEM30A* was mutated in 5–10% of DLBCL, and most of them resulted in loss-of function of TMEM30A [56]. *TMEM30A* encodes the beta-subunit of phospholipid flippase (P4-ATPase), which regulates translocation of phosphatidylserine (PS) from the outer to the inner leaflet of the plasma membrane, maintaining an asymmetric distribution of the phospholipid. TMEM30A is one of the main players regulating the “eat me” signal that promotes phagocytosis of macrophage. TMEM30A loss-of-function enhances cell surface BCR dynamics facilitating more rapid B-cell responses, while it also increases tumor-infiltrating macrophages, suggesting TMEM30A loss of function is associated with a primed microenvironment for phagocytosis [56]. The presence of TMEM30A mutations was a strong favorable prognostic factor in R-CHOP therapy, especially in cases with bi-allelic alterations of TMEM30A. *TMEM30A* mutations were detected as a significant component of BN2 in LymphGen classification and C1 group described by Chapuy et al., both of which are genomic subtypes with favorable outcomes [7,10]. Notably, the authors also demonstrated that cytotoxic treatment (vincristine) or CD47 blockade had a significant therapeutic effect on *TMEM30A*-defficient tumor. These findings provide insight into the roles of “eat me” and “Don’t eat me” signals in DLBCL, enabling the development of novel therapeutic strategies [57]. Figure 1 presents these immune recognition molecules, and the co-stimulatory and co-inhibitory molecules are affected by genetic aberrations.

Genomic aberrations involving epigenetic modifier genes are one of the most frequent events in DLBCL and are found in up to 60% of cases [7,8]. These alterations occur early during the pathogenesis of lymphoma and appear to play a critical role in lymphomagenesis, especially in GCB-DLBCL [58,59]. These epigenetic modifier genes include *Histone-lysine N-methyltransferase 2D (KMT2D)*, *Enhancer of zeste homolog 2 (EZH2)*, *Cyclic-adenosine monophosphate response element-binding protein (CREBBP)*, and *Histone acetyltransferase p300 (EP300)*. Under normal conditions, these epigenetic modifiers control B-cell development and germinal center (GC) formation via histone modification [59]. Gene alterations affecting epigenetic modifiers lead to changes in the expressions of multiple genes—including immune recognition molecules, immune response modifiers, and cytokines—and shape the TME and facilitate lymphoma progression. 

EZH2, a SET domain containing histone methyltransferase, forms part of polycomb repressive complex-2 (PRC2) and is involved in repressing gene expression through the methylation of histone H3 on lysine 27 (H3K27) [60]. Most *EZH2* mutations affect the Y641 residue within the catalytic SET domain, which alters its substrate specificity, leading to a global increase in H3K27me3 [58,61,62]. *EZH2* mutations are enriched in GCB-DLBCL and are the characteristic gene aberrations of EZB-type lymphoma, which harbors both mutations and *BCL2* aberrations [58]. Mutant *EZH2* cooperates with *BCL2* in GC-driven lymphomagenesis and disrupts the *EZH2* normal GC reaction [59]. The normal GC is divided into two anatomical compartments: the dark zone (DZ) and the light zone (LZ). In the normal GC, the DZ primarily comprises a tight cluster of highly proliferative B cells, classically known as “centroblasts”, which undergo rapid proliferation and somatic hypermutation of their immunoglobulin variable genes to generate high-affinity B-cell receptors [63]. The LZ B cells are traditionally referred to as “centrocytes” and interact with follicular dendritic cells (FDC) and TFH, which provide LZ B cells with survival signals. On the other hand, GC B cells with *EZH2* mutation exhibit upregulation of genes involved in interactions with FDC, and show attenuated dependence on TFH cells, and downregulation of genes involved in interactions with TFH, such as CD40 [64,65]. *EZH2* mutation also alters immune recognition molecules expression. DLBCL with *EZH2* mutation exhibits decreased expression of MHC-I and MHC-II, as well as lower levels of CD4^+^ and CD8^+^ T-cell infiltration [29]. *EZH2* mutation has also been reported to repress the expressions of CD58, PRAME, and cancer-testis antigen, and to inhibit anti-tumor effects of T cells and macrophages [66,67]. Anti-EZH2 drugs can restore the expressions of these immune-associated molecules [29]. Since *EZH2* mutation is an inferior prognostic factor, *EZH2* can be regarded as a therapeutic target in DLBCL [29,68]. CREBBP and EP300 are two closely related histone acetyltransferases that display structural and functional similarities, sharing 60% common amino acids, and that function as transcriptional co-activators via H3K27 acetylation [69]. Loss-of-function mutations of *CREBBP* and *EP300* are found in around 25% and 5% of DLBCL, respectively. Their mutations are almost mutually exclusive, suggesting their compensatory function in lymphomagenesis [68]. Studies in several transgenic mouse models have shown that *Crebbp* loss results in downregulation of immune molecules involved in interaction with CD4^+^ T cells, such as *Cd40* and *Ciita* [70,71,72]. Despite the functional similarity between CREBBP and EP300, knock-out of *Ep300* does not lead to downregulation of *Cd40* and *Ciita*. Both *CREBBP* and *EP300* mutations have adverse clinical effects. DLBCL with *CREBBP* mutations shows depletion of tumor-infiltrating CD4^+^ T cells, while *EP300* mutation is associated with M2 macrophage polarization, which induces an immune-suppressive TME [68]. Genes perturbed by *CREBBP* mutations are direct targets of the BCL6/HDAC3 onco-repressor complex. Recently, Mondello et al. reported that HDAC3 inhibitor was able to restore the MHC class II expression in CREBBP-deficient tumor cells, and HDAC3 inhibition represented a novel immune-epigenetic therapy for CREBBP mutant lymphomas [73]. 

KMT2D (MLL2, MLL4) is a SET domain-containing lysine methyltransferase, which mediates H3K4 mono and demethylation (H4K3me1/2) primarily at gene enhancers, resulting in upregulation of target gene transcription [74]. Inactivating mutations of *KMT2D* are one of the most frequent gene alterations in DLBCL, with a reported frequency of around 25% [75]. Combined with BCL2 deregulation, *KMT2D* mutation is sufficient for lymphoma development in vivo [76], recapitulating the co-presence of *KMT2D* mutation and *BCL2* translocation in human DLBCL [8]. KMT2D-deficient B cells exhibit aberrant repression of genes involved in immune signaling, such as CD40, IL10–IL6, and NF-κB signaling [77]. Recently, *KMT2D* mutations were reported to promote tumor-derived TGF-β1 production and to increase regulatory T-cell (Treg) infiltration in tumor tissue, thereby suppressing immunosurveillance [78]. 

## 3. Tumor Microenvironment of DLBCL

Compared with the tumor cell biology of lymphoma, the importance of the microenvironment in which malignant cells arise and subsequently reside has been relatively underestimated [79]. However, recent technological advances enable us to perform an integrative analysis combining genetic aberrations of DLBCL and transcriptional features of non-malignant components. Many studies have noted that T cells in the TME provide lymphoma cells with signals for survival and proliferation. Enrichment of T-cell infiltration was reported to be associated with increased risk of bone marrow or splenic involvement, whereas other studies have found that high T-cell infiltration was associated with significantly better survival [80,81]. Regarding the T-cell subsets, an increased number of Tregs in the TME was associated with longer progression free-survival [82]. In contrast, infiltration with PD-1-expressing T cells was reportedly associated with short PFS and OS [81]. A recent study employing single-cell RNA sequencing revealed that T cells in the TME of B-cell lymphomas, including DLBCLs, could be divided by unsupervised clustering into four categories: conventional T helper cells, TFH, Tregs, and CTL [83]. A computational approach analyzing ligand–receptor interacting pairs revealed that T cells provide multiple regulatory immune response signals, including CD80/CD86–CD28, CD80/CD86–CTLA4, BAFFR–BAFF, and CD40–CD40L. Notably, TFH was the main source of IL-4, which has been indicated to render tumor cell resistance to BTK inhibitors [84,85]. 

The role of macrophages in the TME is somewhat controversial [86]. In the classic model, macrophages are divided into M1 and M2 subtypes, which each have different mechanisms of activation and effector functions [87]. M1 polarization is associated with macrophage-dependent tissue damage and tumor cell killing, whereas M2 polarization is thought to promote tumor growth by down-modulating immunity and favoring angiogenesis [88]. A high density of tissue-associated macrophages (TAM) has been associated with an unfavorable prognosis in DLBCL treated without rituximab [89,90,91,92], whereas high TAM infiltration was linked with a favorable outcome in patients treated with rituximab-containing regimens [91,93]. Notably, a high density of TAM with a M2-like phenotype (represented by CD163 expression) predicted poor prognosis even in patients treated with rituximab [94]. Interestingly, signals derived from TAM (IL-10 and CSF-1) have been reported to enhance macrophage-mediated phagocytosis of rituximab-opsonized lymphoma cells, which may partly explain why rituximab treatment influences the prognostic significance of TAM density [95]. 

Cytokine production of tumor cells or inflammatory cell of TME contributes not only to the proliferation of tumor cells but also to the maintenance of an appropriate environment for the tumor cells [96]. For example, *Interleukin-6* (*IL6*) and *Interleukin-10* (*IL10*) are transcriptionally activated through NF-κB pathway activation in ABC-DLBCL cell lines [97]. IL-6 was originally identified as a T cell–derived cytokine that induced terminal maturation of B cells into plasma cells [98]. Interleukin 10 (IL-10), also originally identified as a helper T-cell product, promotes the proliferation of normal B cells [99]. These cytokines bind to their surface receptors, leading to JAK/STAT pathway activation, cellular proliferation, and further increase in production of IL-6 and IL-10 [97]. Tumor production of these cytokines and serum cytokine levels were significantly correlated [100]. High levels of serum IL-6 and IL-10 have been reported to be poor prognostic factors [101,102,103]. Transforming growth factor-β (TGF-β) has a dual role in tumor suppression and promotion of human malignancies [104]. Loss of the TGF-β antiproliferative response is a hallmark in malignant lymphoid neoplasms [105]. TGF-β/Smad signaling induces expression of S1PR2, a tumor suppressor in DLBCL, and inhibits the activity of an oncogenic transcription factor FOXP1 [106]. The downstream targets of TGF-β, SMAD5 and SMAD1 also act as tumor suppressors, and they are repressed in human DLBCLs [105,107]. High-TGF-β pathway activity was associated with better prognosis in DLBCL [108,109]. However, TGF-β also exerts its tumor promoting effects by inducing migration, invasion, metastasis, angiogenesis, and immune suppression in many types of human malignancies [110]. Recently, Aoki et al. reported that TGF-β production of tumor cells of Lymphocyte-rich classic Hodgkin lymphoma (LRCHL), and the corresponding enrichment of PD-1^+^CXCL13^+^ T cells, may shape the immune microenvironment of LRCHL [111]. Given the clinical, histological and biological similarities with LRCHL, nodular lymphocyte pre-dominant and T-cell/histiocyte-rich large B-cell lymphoma (TCRLBCL), it is plausible that immunosuppressive effect of TGF-β plays an important role in a certain population of DLBCL [112,113,114].

Gene expression profiling based on oligonucleotide microarray analysis has revealed that distinct stroma gene expression signatures have clinical implications. Unsupervised cluster analysis based on gene expression profiles identified a “host response” cluster enriched in an ongoing inflammatory/immune response signature and sharing histological features with T-cell histiocyte-rich large B-cell lymphoma (THRLBCL) [80]. Other studies have revealed that a Stromal-1 signature, featuring extracellular matrix- and histiocyte-associated genes, was associated with a good prognosis, while a Stromal-2 signature, featuring angiogenesis-associated genes was related to a poor prognosis [80,108]. Subsequently, the TME gene expression subclassification was refined by extracting microenvironmental signatures from the transcriptome, which revealed that the DNA methylation patterning of tumor cells was associated with a distinct TME and can be a pharmacological target [109]. Employing next-generation sequencing enabled us to combine the gene expression signatures of the TME with genomic aberrations. For example, the Stromal-1 signature was revealed to be associated with the EZB-MYC-subtype represented by *EZH2* mutations and *BCL2* translocations, which genomically resembles follicular lymphoma [8,12]. This subtype also exhibited a high GC TFH signature, consistent with its similarity to GC light zone cells. In contrast, the EZB-MYC^+^ subtype, which is associated with concurrent *MYC* and *BCL2* rearrangements (so-called “double-hit” lymphomas [DHL]), exhibited a relatively lower GC TFH signature. Accordingly, DHL showed transcriptional similarity with DZ B cells, including low CD4^+^ T-cell infiltration and frequent lack of MHC-I and II expression, suggesting that they harbor strong cell-autonomous survival and proliferation signals, and reduced dependence on the microenvironment [115]. A selectively low T-cell signature was observed in the MCD subtype. Among genetically defined subtypes, the A53 type that features *TP53* inactivation exhibited frequent genomic aberrations affecting *B2M*, enabling escape from immune surveillance and causing lower immune cell infiltration. 

Detailed information on the cellular composition of the TME has only recently been obtained. Steen et al. and Luca et al. have developed Ecotyper, a novel machine-learning framework integrating transcriptome and single-cell RNA sequence, which can yield a discrete set of cellular community networks [12,116]. Ecotyper consists of three key steps: digital purification of cell-type-specific gene expression profiles from bulk tissue transcriptomes, identification and quantitation of transcriptionally defined cell states, and co-assignment of cell states into multicellular communities. Ecotyper has been used to identify 44 transcriptomically defined “cell states” derived from 13 major cell lineages, including five cell states of malignant B cells. For example, state S1 displayed high levels of canonical marker genes associated with GCB DLBCL, whereas states S4 and S5 expressed marker genes of ABC DLBCL. Compared with normal tonsillar B cell phenotypes, S1 showed specificity for germinal center (GC) B cells, S2 and S3 for pre-memory B cells, S4 and S5 for pre-plasmablasts, and S4 for light zone B cells. Each sample was represented as a mixture of cell states, and when tumor samples were classified according to their most abundant B cell state, several states (S2–S4) showing notable representation within and across COO subtypes, while some cell states were enriched in genetic subtypes described by Chapuy et al., or LymphGen subtypes [7,10]. They also identified 39 TME cell states, and the majority of TME cells states including monocytes/macrophages (M1-like) and CD4 T cells (naive) was found to dominate favorable outcomes. In consistent with previous studies, T cells associated with GCB DLBCL were generally deficient in immunomodulatory molecules, while T cell states enriched in ABC DLBCL showed widespread overexpression of co-stimulatory and co-inhibitory molecules including LAG3 and TNFSRF40. This work has also revealed nine multicellular “ecosystems” in DLBCL, constituted by substantially varying numbers of cells belonging to each “cell state”. Compared to previous methods, these ecosystems demonstrated clear improvements in prognostic utility, and the methodology was robust enough to recover ecotypes in RNA-seq data derived from previous studies. For example, lymphoma ecotype (LE) 1 and LE2 were linked to ABC-DLBCL, and LE3 was linked to DHL. In contrast, LE6–8 showed favorable prognosis, and LE8 was enriched for GCB-DLBCL and its related genotypic lesions (EZB, ST2, C3 and DHL). LE6,7,9 were characterized by high stromal component. They also identified that a high content of CD8 T cells expressing CXCR5 was observed in LE5, which predict a greater therapeutic benefit from bortezomib targeting NF-κB signaling. These studies are expanding our understanding of cellular organization in DLBCL, with implications for the development of biomarkers and individualized therapies.

## 4. Systemic Immunity

Aging-related immunological deterioration, i.e., immunosenescence, has long been postulated to be involved in lymphoma development, which is best exemplified by EBV^+^DLBCL [14,15]. EBV^+^DLBCL was first documented as a disease characterized by higher age distribution, aggressive clinical features, and frequent extranodal involvement—such as in the lungs, upper aero-digestive tract, and gastrointestinal tract. Although they express viral-derived antigen, such as LMP1, a significant number of EBV^+^DLBCL cases do not exhibit distinct genomic aberrations associated with immune escape. Only 10–40% of EBV^+^DLBCL have PD-L1 dysregulation, with a slight abundance among nodal lesions [46,117]. In contrast, EBV^+^ large B-cell lymphoma (LBCL) in young patients (age < 45 years) shows frequent (77%) PD-L1 overexpression on tumor cells, preferential nodal involvement, and an excellent prognosis [118]. This difference in PD-L1 positivity rate between the two age groups implies that immunosenescence is involved in EBV-infected tumor cell propagation in elderly patients. 

It has recently been proposed that immunosenescence plays a role in the strong relationship between age and cancer risk. For example, analysis of the immune cell composition of healthy individuals using multidimensional trajectory analysis can reportedly predict all-cause mortality beyond well established risk factors [119]. Furthermore, combining data from immunology and epidemiology, a recent study demonstrated that age-related increases in the incidence of malignant neoplasms can be modeled based on immune system decline due to thymic involution, rather than somatic gene mutation accumulation [120]. Due to thymic involution, T-cell generation during adult life depends on the peripheral proliferation of naïve T cells. The T-cell repertoire of naïve T cells declines with age, whereas the repertoire of memory T cells is maintained [121]. Analysis of circulating memory T cells in peripheral blood has revealed that aged individuals exhibit enrichment of certain T-cell subpopulations, such as GZMK^+^ CD8^+^ T cells and CD4^+^ CTLs [122,123,124]. Immune system depletion in the elderly has been studied especially in cytomegalovirus (CMV)-infected individuals. In such populations, CMV-specific CD8 T cells are shown to be accumulated with age, and to have a decreased immediate effector function [125]. Aside from T-cell subpopulation changes, aging causes genetic and epigenetic changes in T cells. Repetitive replication of T cells due to chronic inflammation can induce telomerase inactivation, leading to telomere shortening and replicative senescence [126,127]. Epigenetic profiling of naïve CD8^+^ T cells from aged individuals revealed that they lose stem-like features and shows similarities with effector T cells [128]. The T cells of elderly individuals may also be affected by age-related clonal hematopoiesis and mosaic chromosomal alterations [129]. The genomic characteristics of EBV^+^DLBCL are frequent gene mutations associated with clonal hematopoiesis recurrently observed in myeloid malignancies [130,131,132]. Notably, angioimmunoblastic T-cell lymphoma (AITL), which is frequently accompanied by abnormal EBV+ B-cell proliferation, harbor somatic mutations of these epigenetic modulator genes such as *TET2* and *DNMT3* [132,133]. Since somatic mutations of epigenetic modulator genes are characteristic of clonal hematopoiesis, it is of questionable value whether genetic changes in T cells are related to defects of inhibiting the proliferation of EBV-infected B cells. In line with this hypothesis, abnormal GC B-cell expansion was observed in the AITL mouse model that lacks *tet2*, and the CD40–CD40L signal provided by *tet2*-deficient TFH was suggested to be involved in this abnormal B-cell expansion [134]. Other T-cell ageing processes include mitochondrial dysfunction and loss of proteostasis [128]. However, it remains unclear how these T-cell ageing processes may lead to impairment of anti-tumor function. Furthermore, analyses of functional and population changes in T cells among aged individuals have been focused on peripheral blood lymphocytes and have not yet elucidate changes of T cells in peripheral tissues. The aging process is also known to be associated with functional changes in tissue-residual immune cells, such as neutrophils, macrophages, and dendritic cells, whereas their associations with cancer risk have not yet been discovered [124]. Unraveling the effects of aging on such immune cells might be helpful in understanding the pathogenesis of immunosenescence-associated lymphomas because these lymphomas frequently arise in extranodal sites [15]. 

## 5. DLBCL in Immune-Privileged Sites

A unique subtype of DLBCL, including primary lymphoma of the central nervous system (PCNSL) and primary testicular lymphoma, exhibits specific molecular features [135]. These organs are regarded as “immune-privileged sites” in which multiple immune-modulating mechanisms exist [136]. Lymphoma cells in these organs frequently harbor immune escape-associated genetic aberrations, including loss of *HLA* loci and *CD58*, and less frequently exhibit *PD-L1/L2* copy number alteration (CNA) and SV [135,136,137]. These molecular features are shared by DLBCL with extranodal site involvement, such as the primary skin, breast, uterus, adrenal, and intravascular system—organs which can be regarded as “relative immune-privileged sites” [10]. Our group has shown that neoplastic PD-L1 expression is present in only a minority of cases, and that cases with neoplastic PD-L1 expression tend to show an aggressive clinical course [136,138,139,140,141]. In particular, PD-L1 expression on tumor cells was found more frequently than in other subtypes of DLBCL due to SV involving the 3ʹ-UTR of *PD-L1*, reaching around 40% of IVL [142,143]. DLBCL in immune-privileged sites is also characterized by frequent presence of *MYD88^L265P^* and/or *CD79B* mutations, which correspond to the MCD subtype in the LymphGen classification algorithm [10]. These mutations can now be detected using cfDNA, which will offer a new diagnostic approach for cases without any available biopsy specimen [143]. *Myd88^L265P^* helps B cells survive without GCB-TFH crosstalk, leading to abnormal autoimmune memory B-cell expansion [144]. *MYD88^L265P^*-bearing lymphoma precursor cells, or pre-tumor cells, have been detected in the peripheral blood of PCNSL patients, indicating that *MYD88^L265P^* is a trunk driver gene mutation of PCNSLs [145]. *MYD88^L265P^* can also be detected in healthy individuals as a consequence of clonal hematopoiesis [146]. A recent study reported that transcriptional features of malignant B cell clones in brain tumors were shared by malignant B cell clones in the cerebrospinal fluid, but not by non-malignant clones in the peripheral blood, suggesting that malignant B cell clones of PCNSL developed within the CNS [147]. Other lymphoma-associated gene mutations can also be detected in autoantibody-producing B cells among autoimmune disease patients [148]. *TBL1XR1* mutations, another driver event of the MCD subtype, induce extranodal lymphomas in a mouse model [149]. *Tblxr1* mutant memory B cells fail to differentiate into plasma cells and, instead, preferentially reenter new GC reactions. Moreover, lymphomas induced by *Tblxr1* mutation showed a high AID activity footprint, recapitulating DLBCL of the MCD subtype. The high immunogenicity of lymphoma cells associated with AID activity may explain the high prevalence of immune escape-associated genetic aberrations among DLBCLs in immune-privileged sites [150]. Immune escape-associated genetic aberrations might help lymphoma precursor cells evade immune surveillance when they enter or reenter the GC of lymph nodes (Figure 2). This notion is further supported by the finding that the breakpoints of *PD-L1* CNA and SV are often within the target motif of AID, which is preferentially expressed in the GC [44,143,151]. It is also possible that *PD-L1* CNA and SV can be acquired after exit from the GC and homing to immune-privileged sites, which is supported by the presence of IVL cases with intratumoral heterogeneous PD-L1 expression across the involved organs [152]. In addition to frequent immune escape-associated genetic aberrations, an immune-tolerogenic TME also contributes to lymphoma development in immune-privileged sites. Although neoplastic PD-L1 expression is very rare in extranodal DLBCL, microenvironmental PD-L1 expression is frequently observed among DLBCL in extranodal lesions, including immune-privileged sites [136,138,139]. More interestingly, microenvironmental PD-L1 expression is a good prognostic indicator for DLBCL in these sites, suggesting that neoplastic cell growth in extranodal lesions might depend on a tolerogenic TME, rather than cell-autonomous proliferation [136,138,139]. A recent study revealed that TME of PCNSL was enriched by T cells with exhausted phenotype represented by high expression of *CD27*, *PDCD1*, *LAG3* and *TNFRSF9* [147]. In addition, some proteins expressed specifically in CNS have been reported to bind to BCR of tumor cells, thereby fostering tumor growth and progression [153,154,155]. The observation that tumor cells of PCNSL display biased IGV rearrangement, with preferential usage of the IGHV4-34 recognizing galectin-3 on various cell types in CNS, which also indicates that antigen-based selection contributes to lymphoma precursor cells expansion [155,156,157,158]. These data strongly support that not only genetic abnormalities of tumor cells but also cellular interactions of tumor cells and TME are indispensable for lymphomagenesis in immune-privileged sites.

The pathogenesis of DLBCL in immune-privileged sites differs between patients with immune deficiency versus immunocompetent patients. Neoplastic cells are frequently infected by EBV, and almost all of them lack immune escape-associated genetic aberrations, *MYD88^L265P^* and/or *CD79B* mutations [159]. They also lack gene mutations affecting *PIM1*, the well known target of AID [160]. Instead, the tumors are accompanied by an immune-tolerogenic TME, enriched with PD-L1-expressing inflammatory cells and TIM-3-positive T cells, showing similarity to DLBCL in immune-privileged sites with microenvironmental PD-L1 expression in immune-competent patients [159]. These findings also exhibit similarities with EBV-positive mucocutaneous ulcer (EBVMCU), which arises in the setting of immunosuppression or immunosenescence. EBVMCU is characterized by mucosal lesions, EBV-positive neoplastic cells lacking PD-L1 expression, and an indolent clinical course [161,162]. Recently, Kawano et al. reported a series of primary adrenal DLBCL, including three out of nine EBV-positive cases with past medical histories of neoplastic disease [138]. These cases raise the possibility that cancer-associated systemic immune changes or immune dysfunction due to chemotherapy contribute to EBV-infected B-cell proliferation in immune-privileged sites, and these findings have also been reported in EBVMCU [163,164]. Although the small and large intestine have not been documented as “immune-privileged” sites, various molecular mechanisms maintaining mucosal immune tolerance have been reported [165]. Ishikawa et al. assessed 62 cases of primary intestinal DLBCL, including ten EBV-positive cases, and almost all EBV-positive cases showed immunosuppressive features, sharing clinicopathological characteristics with PCNSL in immune deficiency. This clearly contrasts with primary gastric EBV-positive DLBCLs, which frequently arise without immunosuppression, and implies that the small and large intestine have biological aspects of an “immune-privileged site” [166,167,168]. Collectively, the pathogenesis of lymphomas in immune-privileged sites appear to be strongly associated with both the immune systems of the involved organs and systemic immunity; however, their biological mechanisms have not yet been well examined.

## 6. Conclusions

As new immune-targeting therapies have been developed, it has become increasingly important to understand the immunological aspects of DLBCL. Moreover, advances of novel methods for analyzing the cell composition of the TME is leading to the discovery of more mechanisms of biological interaction between tumor cells and immune cells. However, there remain many unanswered questions regarding the immunology of DLBCL—for example, relating to the TME of extranodal lesions, underlying mechanisms of immunosenescence and lymphomagenesis, immunological diversity according to ethnicity, and the mechanism of pre-tumor cells homing to extranodal lesions. Further studies are required to expand our current understanding of the immunology of DLBCL to improve the effectiveness of immune-targeting therapies. 

## Figures and Tables

**Figure 1 cancers-15-00835-f001:**
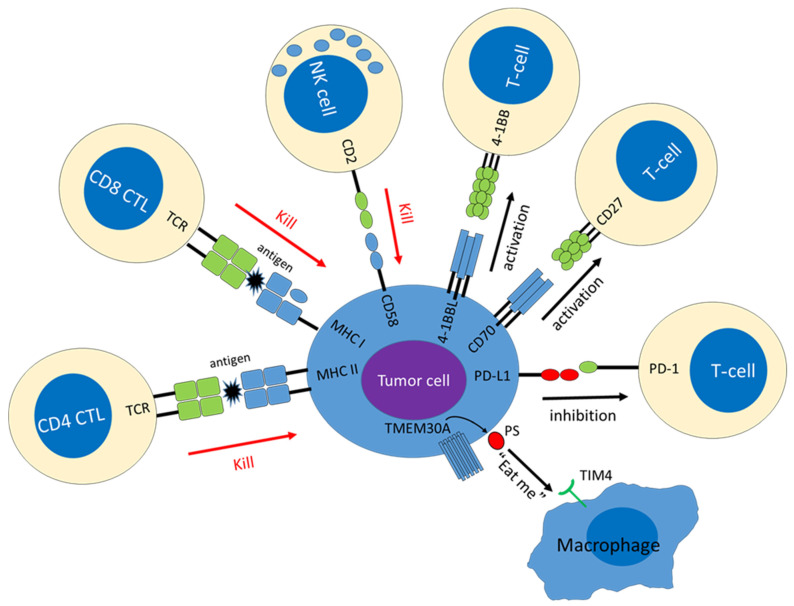
Hallmarks of immune escape mechanisms induced by genetic alterations. Molecules down-regulated in tumor cells are shown in blue, and molecules up-regulated in tumor cells are shown in red.

**Figure 2 cancers-15-00835-f002:**
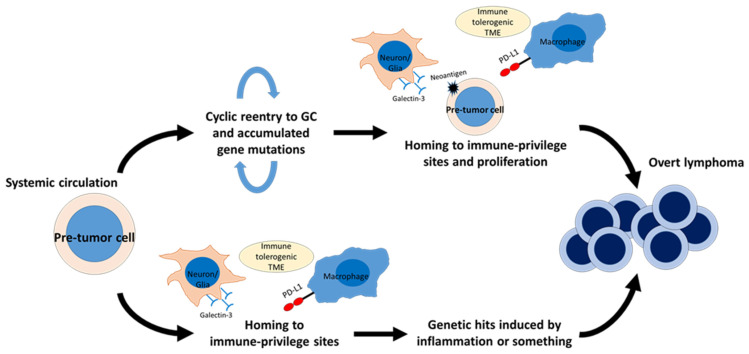
Hypothetical scheme of diffuse large B-cell lymphoma (DLBCL) development in immune-privileged sites. Lymphoma-precursor cells, or “pre-tumor cells”, are thought to be present in systemic circulation. For example, memory B cells harboring *Tbl1xr1* mutation exhibit skewed GC output and an increased tendency to reenter GC. These memory B cells appear to be prone to gaining additional somatic mutations. The immune–tolerogenic TME of immune-privileged sites may help “pre-tumor cells” with high-immunogenicity survive and proliferate, leading to lymphomagenesis (upper side). In DLBCL involving the central nervous system (CNS), “pre-tumor cells” harboring *MYD88^L265P^* are found in peripheral blood, and these cells may gain additional genomic hits after their homing to the CNS (lower side). This model may be applicable to a minority of intravascular large B-cell lymphomas with heterogeneous PD-L1 expression (lower side). Glectin-3 expressed on neurons and glial cells may also contribute to lymphoma precursor cells expansion by antigen-based selection. GC, germinal center; TME, tumor microenvironment.

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
