# Peer review of "The Immunology of DLBCL"

_cancers, 2023, doi:10.3390/cancers15030835_

Round 1
Reviewer 1 Report
Takahara et al. have written a review paper on the immunology of DLBCL. The paper is concise but suffers greatly from a suboptimal lack of logical structure and rather poor English. The introduction is superficial and unfocused. There is not a single histological picture to support the statements of the article. The authors are unable to distinguish between prognostic and predictive markers. The description of "neoantigen burden" needs to be vastly improved.
In its current form the paper does not reach sufficient impact to be published in Cancers (Basel).
Specifically:
1. the authors need to consider the abundant evidence of PTLD and DLBCL in immunosuppressed individuals to support their hypotheses.
2. the authors need to consider a consistent structure for each paragraph, starting with the physiological role of the pathway or marker in question, then summarising the evidence for their dysregulation in DLBCL, and concluding with possible pharmacological interventions.
3. the issue of CD58 mutations is not adequately addressed.
4. the role of CD27 (CD70 ligand) is not addressed.
5. the role of IL6, Il10, TFG-Beta is not addressed.
6. the role of CD47 and the eat-me/don’t eat-me machinery is not addressed.
7. some seminal works e.g. Jiang Y, Ortega-Molina A, Geng H, Ying HY, Hatzi K, Parsa S, McNally D, Wang L, Doane AS, Agirre X, Teater M, Meydan C, Li Z, Poloway D, Wang S, Ennishi D, Scott DW, Stengel KR, Kranz JE, Holson E, Sharma S, Young JW, Chu CS, Roeder RG, Shaknovich R, Hiebert SW, Gascoyne RD, Tam W, Elemento O, Wendel HG, Melnick AM. CREBBP Inactivation Promotes the Development of HDAC3-Dependent Lymphomas. Cancer Discov. 2017 Jan;7(1):38-53. doi: 10.1158/2159-8290.CD-16-0975. Epub 2016 Oct 12. PMID: 27733359; PMCID: PMC5300005, but many others (too many to list) are not considered at all.
8. tumour microenvironment is only treated very superficially. In this regard, I refer for example to Opinto G, Vegliante MC, Negri A, Skrypets T, Loseto G, Pileri SA, Guarini A, Ciavarella S. The Tumor Microenvironment of DLBCL in the Computational Era. Front Oncol. 2020 Mar 31;10:351. doi: 10.3389/fonc.2020.00351. PMID: 32296632; PMCID: PMC7136462. and/or Menter T, Tzankov A. Lymphomas and Their Microenvironment: A Multifaceted Relationship. Pathobiology. 2019;86(5-6):225-236. doi: 10.1159/000502912. Epub 2019 Oct 1. PMID: 31574515.
9. the papers 12 and 92 should be described in a considerably higher detail than in the current paper version.
10. the paragraph on "systemic immunity" should be renamed "immunosenescence”, the role of immune system depletion particularly by CMV should be elaborated, and the most relevant literature considered, for example:
Solana R, Tarazona R, Gayoso I, Lesur O, Dupuis G, Fulop T. Innate immunosenescence: effect of aging on cells and receptors of the innate immune system in humans. Semin Immunol. 2012 Oct;24(5):331-41. doi: 10.1016/j.smim.2012.04.008. Epub 2012 May 4. PMID: 22560929.
Khan N, Hislop A, Gudgeon N, Cobbold M, Khanna R, Nayak L, Rickinson AB, Moss PA. Herpesvirus-specific CD8 T cell immunity in old age: cytomegalovirus impairs the response to a coresident EBV infection. J Immunol. 2004 Dec 15;173(12):7481-9. doi: 10.4049/jimmunol.173.12.7481. PMID: 15585874.
Klapper W, Kreuz M, Kohler CW, Burkhardt B, Szczepanowski M, Salaverria I, Hummel M, Loeffler M, Pellissery S, Woessmann W, Schwänen C, Trümper L, Wessendorf S, Spang R, Hasenclever D, Siebert R; Molecular Mechanisms in Malignant Lymphomas Network Project of the Deutsche Krebshilfe. Patient age at diagnosis is associated with the molecular characteristics of diffuse large B-cell lymphoma. Blood. 2012 Feb 23;119(8):1882-7. doi: 10.1182/blood-2011-10-388470. Epub 2012 Jan 11. PMID: 22238326.
11. a recent work on EBV+ DLBCL bearing CHIP-like mutations published in Blood Advances might be considered: Epub ahead of print. PMID: 36399513.
12. the role of EBV-induced STAT3 signaling and finally PDL1 overexpression deserves more detailed description.
13. the paragraph on lymphomas at immunoprivileged sites contains some controversial statements, especially about PDL1, which need to be revised.
In general:
Professional English proofreading is required.
Some in situ (histopathological) evidence of the relevant changes - respecting the scope of the paper - in DLBCL in the form of images might be considered.
Author Response
1: The authors need to consider the abundant evidence of PTLD and DLBCL in immunosuppressed individuals to support their hypotheses.
⇒ The reviewer has raised an important issue to consider. However, these LPD in immunosuppressed individuals contains reactive lesions, and treated separately from Large-B cell lymphomas in 5th edition of WHO classification. Therefore, we omitted the description about these entities.
2: The authors need to consider a consistent structure for each paragraph, starting with the physiological role of the pathway or marker in question, then summarising the evidence for their dysregulation in DLBCL, and concluding with possible pharmacological interventions.
⇒ We thank the reviewer for bringing this to our attention. We discussed the genetic aberrations in order of frequency in the section of genetic aberrations (section #2). We start with the frequency of each genetic events, subsequently discuss functions in normal cells, functional roles in lymphoma. In the section of TME (section #3), we start with each component of TME (ie, T cell, macrophage, cytokines) and discuss the integrative analysis of TME in the latter part of section #3. We did not discussed the normal physiological roles in the section of immunosenescence and DLBCL in immune privileged sites (section #4 and #5). We also added the descriptions of future prospect of pharmacological interventions about “Ecotyper”, “Don’t eat me signal”, CREBBP and EP300.
(Page 11, line 3 to line 6)
Genes perturbed by CREBBP mutations are direct targets of the BCL6/HDAC3 onco-repressor complex. Recently, Mondello et al. reported that HDAC3 inhibitor was able to restore the MHC class II expression in CREBBP-deficient tumor cells, and HDAC3 inhibition represented a novel immune-epigenetic therapy for CREBBP mutant lymphomas [31915197]
3: The issue of CD58 mutations is not adequately addressed.
⇒ We thank the reviewer for bringing this to our attention. We added the sentences in the section about CD58 as follows: (Page 5, line 3). The modified manuscript is written in red.
CD58 expression on tumor cells is required both for NK cell and CTL-mediated cytolysis of DLBCL. Inhibiting CD58 results in diminished recognition and cytolysis of target cells by both CTLs and NK cells, and re-expression of CD58 induced a significant increase in NK cell-mediated cytolysis [PMID: 22137796]
4: The role of CD27 (CD70 ligand) is not addressed.
⇒ We discuss CD27 in Page 6, line 3 to line 12.
5: The role of IL6, Il10, TFG-Beta is not addressed.
⇒ The reviewer has raised an important issue to consider. We added the sentences about these cytokines in the section of TME, page 13 line 3 to page 14, line 4.
Cytokine production of tumor cells or inflammatory cell of TME contributes not only to the proliferation of tumor cells but also to the maintenance of an appropriate environment for the tumor cells [19017177]. For example, Interleukin-6 (IL6) and Interleukin-10 (IL10) are transcriptionally activated through NF-κB pathway activation in ABC-DLBCL cell lines [18160665]. IL-6 was originally identified as a T cell–derived cytokine that induced terminal maturation of B cells into plasma cells [3491322]. Interleukin 10 (IL-10), also originally identified as a helper T-cell product, promotes the proliferation of normal B cells [1371884]. These cytokines bind to their surface receptors, leading to JAK/STAT pathway activation, cellular proliferation, and further increase of production of IL-6 and IL-10 [18160665]. Tumor production of these cytokines and serum cytokine levels were significantly correlated [8968107]. High-level of serum IL-6 and IL-10 have been reported to be poor prognostic factors [8400266, 22323454, 21902578]. Transforming growth factor-β (TGF-β) has a dual role in tumor suppression and promotion of human malignancies [33409254]. Loss of the TGF-β antiproliferative response is a hallmark in NHLs [23955273]. TGF-β/Smad signaling induces expression of S1PR2, a tumor suppressor in DLBCL, and inhibits the activity of oncogenic transcription factor FOXP1 [29615404]. The downstream targets of TGF-β, SMAD5 and SMAD1 also act as a tumor suppressor, and they are repressed in human DLBCLs [20133617, 23955273]. High- TGF-β pathway activity was associated with better prognosis in DLBCL [19038878, 33541860]. However, TGF-β also exerts its tumor promoting effects by inducing migration, invasion, metastasis, angiogenesis and immune suppression in many types of human malignancies [30060514]. Recently, Aoki et al. reported that TGF-β production of tumor cells of Lymphocyte-rich classic Hodgkin lymphoma (LRCHL), and the corresponding enrichment of PD-1+CXCL13+ T cells, may shape the immune microenvironment of LRCHL [34615710]. Given the clinical, histological and biological similarities with LRCHL, nodular lymphocyte pre-dominant and T-cell/histiocyte-rich large B-cell lymphoma (TCRLBCL), it is plausible that immunosuppressive effect of TGF-β plays an important role in a certain population of DLBCL [26658840, 30213827, 14508396].
6: The role of CD47 and the eat-me/don’t eat-me machinery is not addressed.
⇒ The reviewer raised an important issue to address. We added the sentence of eat-me/don’t eat-me machinery as follows. (Page 8, line 9 to Page9, line 3)
Recently, Ennishi et al. discovered TMEM30A mutation was mutated in 5-10-% of DLBCL, and most of them resulting in loss-of function of TMEM30A [32094924]. TMEM30A encodes the beta-subunit of phospholipid flippase (P4-ATPase) and a component of flippase, which regulates translocation of phosphatidylserine (PS) from the outer to the inner leaflet of the plasma membrane, maintaining an asymmetric distribution of the phospholipid. TMEM30A is one of the main players regulating the “eat me” signal that promotes phagocytosis of macrophage. TMEM30A loss-of-function enhances cell surface BCR dynamics facilitating more rapid B-cell responses, while it also increase tumor-infiltrating macrophages, suggesting TMEM30A loss of function is associated with a primed microenvironment for phagocytosis [32094924]. TMEM30A mutation is a strong favorable prognostic factor in R-CHOP therapy, especially in cases with bi-allelic alterations of TMEM30A. TMEM30A mutations were detected as a significant component of BN2 in LymphGen classification and C1 group described by Chapuy et al., both of which are genomic subtypes with favorable outcomes []. Notably, the authors also demonstrated that cytotoxic treatment (vincristine) or CD47 blockade had a significant therapeutic effect on TMEM30A-defficient tumor. These findings provide insight into the roles of “eat me” and “Don’t eat me” signals in DLBCL, enabling the development of novel therapeutic strategies [34511583].
7: Some seminal works e.g. Jiang Y, Ortega-Molina A, Geng H, Ying HY, Hatzi K, Parsa S, McNally D, Wang L, Doane AS, Agirre X, Teater M, Meydan C, Li Z, Poloway D, Wang S, Ennishi D, Scott DW, Stengel KR, Kranz JE, Holson E, Sharma S, Young JW, Chu CS, Roeder RG, Shaknovich R, Hiebert SW, Gascoyne RD, Tam W, Elemento O, Wendel HG, Melnick AM. CREBBP Inactivation Promotes the Development of HDAC3-Dependent Lymphomas. Cancer Discov. 2017 Jan;7(1):38-53. doi: 10.1158/2159-8290.CD-16-0975. Epub 2016 Oct 12. PMID: 27733359; PMCID: PMC5300005, but many others (too many to list) are not considered at all.
⇒ We added the some review works to the reference.
8: Tumour microenvironment is only treated very superficially. In this regard, I refer for example to Opinto G, Vegliante MC, Negri A, Skrypets T, Loseto G, Pileri SA, Guarini A, Ciavarella S. The Tumor Microenvironment of DLBCL in the Computational Era. Front Oncol. 2020 Mar 31;10:351. doi: 10.3389/fonc.2020.00351. PMID: 32296632; PMCID: PMC7136462. and/or Menter T, Tzankov A. Lymphomas and Their Microenvironment: A Multifaceted Relationship. Pathobiology. 2019;86(5-6):225-236. doi: 10.1159/000502912. Epub 2019 Oct 1. PMID: 31574515.
⇒ We added the description about cytokines in section #3 and TME of DLBCL in immune-privileged sites (section #5) as reply of Reviewer’s comment #5 and #13.
9: The papers 12 and 92 should be described in a considerably higher detail than in the current paper version.
⇒ We added the detailed description about these studies of “ecotype” as follows. (Page 15, line 9 to Page 16, line 12)
Ecotyper consist of three key steps: digital purification of cell-type-specific gene expression profiles from bulk tissue transcriptomes, identification and quantitation of transcriptionally defined cell states, and co-assignment of cell states into multicellular communities. Ecotyper has been used to identify 44 transcriptomically defined “cell states” derived from 13 major cell lineages, including five cell states of malignant B cells. For example, state S1 displayed high levels of canonical marker genes associated with GCB DLBCL, whereas states S4 and S5 expressed marker genes of ABC DLBCL. Compared with normal tonsillar B cell phenotypes, S1 showed specificity for germinal center (GC) B cells, S2 and S3 for pre-memory B cells, S4 and S5 for pre-plasmablasts, and S4 for light zone B cells. Each sample was represented as a mixture of cell states, and when tumor samples were classified according to their most abundant B cell state, several states (S2–S4) showing notable representation within and across COO subtypes, while some cell states were enriched in genetic subtypes described by Chapuy et al., or LymphGen subtypes. They also identified 39 TME cell states, and the majority of TME cells states including monocytes/macrophages (M1-like) and CD4 T cells (naive) was found to dominate favorable outcomes. In consistent with previous studies, T cells associated with GCB DLBCL were generally deficient in immunomodulatory molecules, while T cell states enriched in ABC DLBCL showed widespread overexpression of co-stimulatory and co-inhibitory molecules including LAG3 and TNFSRF40. This work has also revealed nine multicellular “ecosystems” in DLBCL, constituted by substantially varying numbers of cells belonging to each “cell state”. Compared to previous methods, these ecosystems demonstrated clear improvements in prognostic utility, and the methodology was robust enough to recover ecotypes in RNA-seq data derived from previous studies. For example, lymphoma ecotype (LE) 1 and LE2 were linked to ABC-DLBCL, and LE3 was linked to DHIT. In contrast, LE6-8 showed favorable prognosis and LE8 was enriched for GCB-DLBCL and its related genotypic lesions (EZB, ST2, C3 and DHIT). LE6,7,9 were characterized by high stromal component. They also identified that a high content of CD8 T cell state expressing CXCR5 was observed in LE5, which predict a greater therapeutic benefit from bortezomib targeting NF-κB signaling.
10: The paragraph on "systemic immunity" should be renamed "immunosenescence”, the role of immune system depletion particularly by CMV should be elaborated, and the most relevant literature considered, for example …Solana R, Tarazona R, Gayoso I, Lesur O, Dupuis G, Fulop T. Innate immunosenescence: effect of aging on cells and receptors of the innate immune system in humans. Semin Immunol. 2012 Oct;24(5):331-41. doi: 10.1016/j.smim.2012.04.008. Epub 2012 May 4. PMID: 22560929.
Khan N, Hislop A, Gudgeon N, Cobbold M, Khanna R, Nayak L, Rickinson AB, Moss PA. Herpesvirus-specific CD8 T cell immunity in old age: cytomegalovirus impairs the response to a coresident EBV infection. J Immunol. 2004 Dec 15;173(12):7481-9. doi: 10.4049/jimmunol.173.12.7481. PMID: 15585874.
Klapper W, Kreuz M, Kohler CW, Burkhardt B, Szczepanowski M, Salaverria I, Hummel M, Loeffler M, Pellissery S, Woessmann W, Schwänen C, Trümper L, Wessendorf S, Spang R, Hasenclever D, Siebert R; Molecular Mechanisms in Malignant Lymphomas Network Project of the Deutsche Krebshilfe. Patient age at diagnosis is associated with the molecular characteristics of diffuse large B-cell lymphoma. Blood. 2012 Feb 23;119(8):1882-7. doi: 10.1182/blood-2011-10-388470. Epub 2012 Jan 11. PMID: 22238326.
⇒ The reviewer raised an important issue to address. We added the sentence of CMV infection and deterioration of T cell functions as follows. (Page 17, line 16 to line 18)
Immune system depletion in the elderly has been studied especially in cytomegalovirus (CMV)-infected individuals. In such populations, CMV-specific CD8 T cells are shown to be accumulated with age, and they have a decreased immediate effector function. Furthermore, infection with CMV also reduces immunity to other pathogens such as EBV [15585874].
11: A recent work on EBV+ DLBCL bearing CHIP-like mutations published in Blood Advances might be considered: Epub ahead of print. PMID: 36399513.
⇒ This paper suggested by the reviewer describe that genetic aberrations of EBV+ DLBCL associated with myeloid CHIP, rather with lymphoid CHIP. As the reviewer’s suggestion, we changed the sentence of EBV+ DLBCL as follows (Page 18, line 4 to line 7).
The genomic characteristics of EBV+DLBCL is frequent gene mutations associated with clonal hematopoiesis recurrently observed in myeloid malignancies [34663986, 36399513, 30683910]. Notably, angioimmunoblastic T-cell lymphoma (AITL), which are frequently accompanied by abnormal EBV+ B-cell proliferation, harbor somatic mutations of these epigenetic modulator genes such as TET2 and DNMT3.
12: The role of EBV-induced STAT3 signaling and finally PDL1 overexpression deserves more detailed description.
⇒ The reviewer raised an important issue to address. We added the sentence of STAT3 activation as a mechanism of PD-L1 overexpression as follows. (Page 8 line 1 to Page 8, line 4)
In addition, several intracellular signaling pathways are known to induce PD-L1 expression [31668929]. For example, STAT3 binds to PD-L1 gene promoter, and is required for PD-L1 gene expression [19088198]. The role of STAT3 activation in lymphoma has been highlighted mainly in the ABC DLBCL and EBV+DLBCL, and EBV oncoprotein LMP1 can induce STAT3 activation [21307189, 30054295].
13: The paragraph on lymphomas at immunoprivileged sites contains some controversial statements, especially about PDL1, which need to be revised.
⇒ The reviewer raised an important issue to address. Although our descriptions are oriented to PD-L1, recent studies on TME of PCNSL have revealed the important roles of TME in lymphomagenesis. We added the sentence in the section of section 5 “DLBCL in immune-privileged sites” as follows. (Page 19, line 19 to Page 20, line 2) (Page 21, line 1 to line 9) (Figure legend 2)
A recent study reported that transcriptional features of malignant B cell clones in brain tumors were shared by malignant B cell clones the CSF, but not by non-malignant clones in the peripheral blood, suggesting that malignant B cell clones of PCNSL developed within the CNS [36153593]. Other lymphoma-associated gene…
…More interestingly, microenvironmental PD-L1 expression is a good prognostic indicator for DLBCL in these sites, suggesting that neoplastic cell growth in extranodal lesions might depend on a tolerogenic TME, rather than cell-autonomous proliferation [109,111,112]. A recent study revealed that TME of PCNSL were enriched by T cells with exhausted phenotype represented by high expression of CD27, PDCD1, LAG3 and TNFRSF9 [36153593]. In addition, some proteins expressed specifically in CNS are reported to bind to BCR of tumor cells, thereby fostering tumor growth and progression [32193251, 30249786, 26116512]. The observations that tumor cells of PCNSL display biased IGV rearrangement, with preferential usage of the IGHV4-34 recognizing galectin-3 on various cell types in CNS, also indicate antigen-based selection contribute to lymphoma precursor cells expansion [26116512, 10595937, 10477699, 25383641]. These data strongly supports that not only genetic abnormalities of tumor cells but also cellular interactions of tumor cells and TME are indispensable for lymphomagenesis in immune-privileged sites.
Figure legend 2
Glectin-3 expressed on Neurons and glial cells may also contribute to lymphoma precursor cells expansion by antigen-based selection. GC, germinal center; TME, tumor microenvironment.
General comments;
Professional English proofreading is required.
Some in situ (histopathological) evidence of the relevant changes - respecting the scope of the paper - in DLBCL in the form of images might be considered.
⇒ Although English proofreading is required, we have our manuscript edited by professional English editing service (SanFrancisco edit). If more extensive English proofreading and histological images are necessary, another few days would be required.

Reviewer 2 Report
Consider these minor revisions
Page 2, L 70. MCD: not explained before. Explain the meaning of the initials
Page 5, L 198: lymphagenesis?
Page 7, L289 consider abbreviating double hit lymphomas as DHL
Page 9, L368: explain CNA
Figure 1: consider broader explanation explaining the mechanism in disease

Author Response
Page 2, L 70. MCD: not explained before. Explain the meaning of the initials
Page 5, L 198: lymphagenesis?
Page 7, L289 consider abbreviating double hit lymphomas as DHL
Page 9, L368: explain CNA
⇒ We thank the reviewer very much for bringing these to our attention. We have made the necessary corrections.
Figure 1: consider broader explanation explaining the mechanism in disease
⇒ As the reviewer’s suggestion, we added the description of “Eat me signal” to Figure 1.

Reviewer 3 Report
The authors present a comprehensive review on the immunology of DLBCL. Addition of a section on potential therapeutic targets, published and ongoing clinical trials will add greater clinical impact to the manuscript. Implications on selecting and outcomes post-CAR T or BiTE therapy for specific genetic/immunologic subsets could be discussed. Minor spell check and syntax edits are needed. Overall a well written review incorporating all key elements from published literature.
Author Response
The authors present a comprehensive review on the immunology of DLBCL. Addition of a section on potential therapeutic targets, published and ongoing clinical trials will add greater clinical impact to the manuscript. Implications on selecting and outcomes post-CAR T or BiTE therapy for specific genetic/immunologic subsets could be discussed. Minor spell check and syntax edits are needed. Overall a well written review incorporating all key elements from published literature.
⇒ The reviewer raised an important issue to address. Although it is difficult to address all of the reviewer’s concern in ten days, we added the description about dysregulated mechanisms of DLBCL which can be therapeutic targets, such as, “Don’t eat me signal”, CREBBP and EP300.
